# Use of Spinal Anaesthesia with Anaesthetic Block of Intercostal Nerves Compared to a Continuous Infusion of Sufentanyl to Improve Analgesia in Cats Undergoing Unilateral Mastectomy

**DOI:** 10.3390/ani11030887

**Published:** 2021-03-20

**Authors:** Vincenzo Cicirelli, Pasquale Debidda, Nicola Maggio, Michele Caira, Daniela Mrenoshki, Giulio G. Aiudi, Giovanni M. Lacalandra

**Affiliations:** Department of Veterinary Medicine, University of Bari Aldo Moro, 70121 Bari, Italy; pdebidda1975@gmail.com (P.D.); dottormaggio@gmail.com (N.M.); michele.caira@uniba.it (M.C.); daniela.mrenoshki@uniba.it (D.M.); giulioguido.aiudi@uniba.it (G.G.A.); giovannimichele.lacalandra@uniba.it (G.M.L.)

**Keywords:** pain management, loco-regional anaesthesia, levobupivacaine, subarachnoid anaesthesia, intercostal nerve block

## Abstract

**Simple Summary:**

Postoperative analgesia is very important because pain causes various negative effects that prevent patient recovery. Our study aimed to evaluate the analgesic efficacy of subarachnoid anaesthesia combined with intercostal nerve block compared with a constant-rate infusion of sufentanyl citrate in cats undergoing unilateral mastectomy. This study demonstrated that the use of spinal anaesthesia with anaesthetic block of intercostal nerves, using levobupivacaine, guarantees long-lasting and high-quality analgesic coverage and minimises the post-surgical pain inevitably associated with invasive surgical procedures such as radical mastectomy. This study stemmed from a general trend towards increasing attention on postoperative pain after spaying procedures in cats. Since veterinarians are becoming more focused on relieving surgical pain, anaesthetists are expected to use better protocols that can minimise pain and therefore optimise surgical results. This method, considering the relative simplicity of its execution, can be used in daily clinical practice.

**Abstract:**

Unilateral mastectomy is a common surgical procedure in feline species and requires postoperative pain management. Our study aimed to evaluate the analgesic efficacy of subarachnoid anaesthesia combined with an intercostal nerve block, in comparison with the use of sufentanyl citrate administered as a constant-rate infusion (CRI). Twenty cats were randomly divided into two groups (*n* = 10/group) based on the analgesic protocol used: the first received loco-regional anaesthesia with levobupivacaine (LR group), and the second received a CRI of sufentanyl (SUF group). The evaluation criteria during surgery were the need for a bolus of fentanyl in the event of an increased heart rate or increased blood pressure. In the postoperative period, the levels of comfort/discomfort and pain were used to obtain a score according to the UNESP-Botucatu multimodal scale. Subjects who scored above seven received analgesic drug supplementation. Intraoperative analgesia was satisfactory, with good haemodynamic stability in both groups. Four patients in the LR group required an extra dose of methadone after they achieved the sternal decubitus position, whereas those in the SUF group required many more doses. The analgesia achieved in the LR group was more satisfactory than that in the SUF group.

## 1. Introduction

Mammary gland tumours account for 17% of all tumours in cats, and most of them are malignant and fast-growing [1]. Therefore, unilateral radical mastectomy is recommended for the treatment of mammary tumours [2]. However, this type of surgery produces severe postoperative pain in cats because it is an extensive and invasive surgery that results in inflammation, oedema, and massive nociceptive stimulation [3]. Good analgesia is very important because pain causes various deleterious effects that hinder patient recovery. In fact, an animal with postoperative pain is more likely to have poor anaesthetic recovery, a decreased appetite, slower wound healing, and increased anxiety [4]. The use of local anaesthetics added to general anaesthesia may help to control postoperative pain, constituting a multimodal analgesic strategy. Local anaesthetic can be injected into the operation site or infiltrated into the wound, inducing a rapid onset analgesic effect [1]. Furthermore, tumescent local anaesthesia using a large volume of diluted local anaesthetic for extensive subcutaneous tissue infiltration has been used for mastectomy in cats, providing effective analgesia during veterinary surgery and in the postoperative period [5]. In addition, spinal anaesthesia is an excellent regional technique, which results in altered nerve function by deposition of local anaesthetics in the subarachnoid space [6]. Of the regional anaesthetic techniques, spinal anaesthesia shows some excellent characteristics, such as a good quality of nerve block and fewer side effects, compared with other regional techniques [7]. Local anaesthetic drugs used in spinal anaesthesia can completely block the transmission of pain (in conscious patients) or nociceptive signals (in anaesthetised patients), thereby resulting in profound analgesia and low incidence of dose-related adverse effects [8]. However, there are few studies on spinal anaesthesia in cats undergoing unilateral mastectomy [5,6,7]. In this study, in addition to spinal administration, we used the intercostal blockade. This block, usually used for patients with rib fractures or after thoracotomy or for pleural drainage, was used in this study as an aid to the spinal cord to further decrease the pain of unilateral mastectomy [9]. The advantage is that this blockage eliminates input and desensitises the tissues innervated by the intercostal nerves located between each rib, providing analgesia without respiratory depression [9]. To assess pain in veterinary patients, several multiparametric pain scales have been validated and used worldwide. The UNESP-Botucatu multimodal scale, developed using a psychometric methodology, takes into account postural attitudes and facial expressions, vocalisations and mood, the response to manipulation and palpation of the painful area, physical activity and gait, and the subject’s behaviour towards the surrounding environment [10]. This scale was considered the most appropriate for the type of evaluation performed in this study. This study aimed to evaluate the analgesic efficacy of a loco-regional anaesthesiology procedure, namely, subarachnoid anaesthesia combined with intercostal nerve block using levobupivacaine, compared with the use of sufentanyl citrate administered as a constant-rate infusion in cats undergoing unilateral mastectomy.

## 2. Materials and Methods

### 2.1. Pre-Surgery Procedure

The study was performed in the Obstetrics, Gynaecological, and Andrological Clinic section of the Veterinary Hospital of the Veterinary Medicine Department, University of Bari Aldo Moro. This study involved the recruitment of 20 cats aged 11.2 to 16.8 years, weighing 2.8 to 4 kg, without previous pathologies, and classified as a low aesthetic risk class (ASA 2). Cats were selected for elective unilateral mastectomy. After general examinations, all the cats had a thoracic radiograph, abdominal ultrasound scan, and routine blood tests (complete blood count and metabolic panel), and their mammary gland tumours were staged according to the tumour node metastasis (TNM) classification system [11]. Informed consent was obtained from the owners before the experiment. The cats were randomly assigned to two groups (*n* = 10/group) based on the analgesic protocol used: the first group received L3/L4 spinal anaesthesia with anaesthetic block of intercostal nerves with levobupivacaine (Levobupivacaina Kabi^®^; Fresenius Kabi, Verona, Italia S.r.l.) (LR group), and the second group received systemic analgesia with sufentanyl (Sufentanyl Hameln^®^; Tillomed, Milano, Italia S.r.l.) through a loading bolus followed by infusion at a constant rate (SUF group). In both groups, patients received pre-medication with alfaxalone (Alfaxan^®^; Dechra Veterinary Products Srl, Torino, Italy) at a dosage of 5 µg/kg, and methadone hydrochloride (Semfortan^®^, Dechra Veterinary Products S.r.l., Torino, Italy) at a dosage of 0.25 mg/kg mixed in the same syringe and injected intramuscularly. When the sedative effect was achieved, a 24 G venous catheter was inserted into the cephalic vein to start standard maintenance fluid therapy, and if necessary, we quickly injected the necessary drugs. This was followed by the administration of 100% oxygen via a face mask using a Mapleson F circuit. Induction of anaesthesia (common in both groups) was achieved by intravenous administration of alfaxalone at a dosage of 2 mg/kg. When the inductive effect was achieved, the patients were intubated and connected to the anaesthesia trolley using a Mapleson F respiratory circuit. Maintenance of the hypnotic state during the procedure was ensured by the administration of sevoflurane (EtSev 2.5%, SevoFlo^®^, Ecuphar Italia S.r.l., Milano, Italy) through the respiratory circuit. From this point, and continuously throughout the surgery, instrumental monitoring of the following vital parameters was performed (pre-incision values): heart rate, electrocardiographic trace, pulse oximetry, carbon dioxide concentration at the end of expiration, percentage of sevoflurane at the end of expiration, and non-invasive blood pressure and body temperature (GE-Datex Ohmeda Carestation 620 Anaesthesia Cart, GE-Datex Ohmeda B 450 Monitor, Milano, Italy).

### 2.2. Spinal Anaesthesia with an Anaesthetic Block of Intercostal Nerves

At this point, for cats belonging to the LR group, loco-regional anaesthesia techniques were performed. Spinal (or subarachnoid) anaesthesia was first performed at the level of the L3/L4 space. The patient was placed in the left lateral decubitus position with the hind limbs flexed and directed toward the abdomen, as far as possible, in a cranial direction. This was necessary to flex the spine and slightly widen the lumbar inter-arch spaces to facilitate the introduction of the spinal needle (Atraucan Paed, 26 G, length 50 mm; B. Braun, Milano, Italy) at the level of the desired space. A paramedian approach was chosen to reach the subarachnoid space, inserting the needle tip laterally with respect to the midline, 1 cm caudally and laterally from the interspinous space of interest. The needle was inserted at an angle of approximately 15° with respect to the sagittal plane and in the craniomedial direction. Once the liquor was released, the needle was connected to a syringe filled with a 0.5% levobupivacaine solution. The dosage of the anaesthetic used was 0.35 mg/kg of 0.5% levobupivacaine, with total volumes ranging from 0.20 to 0.30 mL based on patient weight. Subsequently, an intercostal nerve block from T12 to T4 was performed, using a 0.5% levobupivacaine solution at a dosage of 0.05 mL/kg per point (volume injected for each intercostal space ranging from 0.14 to 0.20 mL depending on patient weight). With the patient positioned in the lateral decubitus position and after aseptic preparation of the surgical area, the lateral aspect of each rib was identified at the level of the proximal third with a finger of the non-dominant hand. At this point, a 25-gauge, 25-mm needle (connected to a 1 mL syringe previously filled with a 0.5% levobupivacaine solution) was inserted through the skin until it reached the caudo-lateral costal margin. Immediately after the tip of the needle crossed the caudal margin of the rib due to a gentle redirection in the cranio-medial direction, the caudo-medial margin of the rib was reached, in correspondence with the vein, artery, and intercostal nerve. The anaesthetic solution was injected at this site, and the procedure was repeated in the intercostal spaces from T12 to T4. For the cats in the SUF group, systemic analgesia was performed with administration of sufentanyl citrate (50 µg/mL) with a bolus loading of 1 µg/kg and maintenance of 1.5 µg/kg/h at a constant-rate intravenous infusion until the end of the surgery.

### 2.3. Surgical Procedure

The skin of the area affected by the surgical procedure was then shaved and aseptically prepared, and the animals were placed in the dorsal decubitus position. Throughout the procedure, measures were taken to minimise heat dispersion, mainly by positioning the patient on an air-heated mat (Bair Hugger model 505; 3M, USA). All operations were performed by the same surgeon and by the same operating team in full compliance with the leges artis [12] for a single-sided mastectomy and lasted on average 50 min (SD 5 min). In the event of an increased heart rate or blood pressure (>30% compared to pre-incision values) during the procedure in response to surgical pain in both groups, a bolus of fentanyl was administered [13] intravenously at 2 µg/kg (Fentadon^®^; Eurovet Animal Health BV, Bladel, The Netherlands).

In the immediate postoperative period, all cats were administered subcutaneous meloxicam at 0.20 mg/kg (Metacam, Virbac, Milan, Italy). The evaluation criteria during surgery were haemodynamic stability and the need for additional pain medication.

### 2.4. Post-Surgery Pain Evaluation

In the postoperative period, the level of comfort/discomfort and pain was assessed according to the UNESP-Botucatu multimodal scale [13,14], which assigns a score of 0 (comfort or no pain) to 30 (maximum pain). Postoperative observations began when the subjects achieved the sternal decubitus position (0 h) and were repeated at 1, 2, 4, and 8 h. Subjects who scored above 7 received analgesic drug supplementation—methadone hydrochloride at a dosage of 0.3 mg/kg injected intramuscularly. The UNESP-Botucatu scores were used to highlight the differences between the two groups.

### 2.5. Ethics

This study was performed in accordance with the ethical guidelines of the Animal Welfare Committee. Institutional Review Board approval of the study was obtained from the University of Bari Aldo Moro with approval number 14/19 (21/10/2019). Animal procedures were performed following the Directive 2010/63/Eu of the European Parliament (Italian DL 26/2014).

### 2.6. Data Analysis

Compiled data were entered into a database created with an Excel spreadsheet and analysed using Stata MP16 software.

Continuous variables are described as means ± standard deviations and ranges, and categorical variables as proportions. Skewness and kurtosis tests were used to evaluate the normality of continuous variables, and a normalisation model was constructed to normalise those not normally distributed. The Student’s *t*-test for independent data was used to compare continuous variables between groups, the analysis of variance (ANOVA) for repeated measures was used to compare continuous variables between groups, and the χ^2^ and exact Fisher’s tests were used to compare proportions.

To assess the determinants of fentanyl and methadone administration, two separate multivariate logistic regression models were built, considering administration as an outcome, and body weight (kg), age at enrolment (years), and group (LR vs. SUF) as determinants; and adjusted odds ratios (ORs) with 95% confidence intervals (CIs) were calculated.

To assess the differences in UNESP-Botucatu scores between hours 8 and 0, a multivariate linear regression model was built, considering the difference as outcome and body weight (kg), age at enrolment (years), and group (LR vs. SUF) as determinants, and the coefficients of correlation with 95% CIs were calculated.

For all tests, a two-sided *p*-value <0.05 was considered statistically significant.

## 3. Results

The study sample consisted of 20 female cats: 10 (50.0%) in the LR group and 10 (50.0%) in the SUF group. The characteristics of the sample, by group, are described in Table 1.

No significant differences were observed in intraoperative fentanyl administration between the LR group (20.0%; *n* = 2/10) and the SUF group (40.0%; *n* = 4/10; *p* = 0.628). Each subject was administered a 0.5 mg/kg bolus of the drug.

ANOVA for repeated measures showed significant differences between groups in the UNESP-Botucatu scores (*p* < 0.0001), detection times (*p* < 0.0001), and interaction between time and group (*p* < 0.001; Figure 1).

The observed events related to methadone administration are described in Table 2.

Multivariate analysis showed a statistically significant association between fentanyl administration and cat age (aOR = 0.2; 95% CI = 0.1–0.9; Table 3); no further associations were observed between outcomes and determinants (*p* > 0.05; Table 3, Table 4 and Table 5).

## 4. Discussion

This study stems from a general trend towards a greater focus on postoperative pain after procedures that have left cats with no analgesic coverage for surgical pain for a long time [15]. As veterinary surgeons have become more focused on relieving surgical pain, anaesthesiologists are using better protocols that can minimise pain and thus optimise surgical outcomes [16].

The team of veterinarians involved in this study ensured that the utmost attention was paid to optimal analgesia, both intraoperatively and postoperatively. In fact, the results of this study showed that all patients underwent mastectomy with a good anaesthetic and analgesic technique.

The anaesthesiologists were careful while administering fentanyl intraoperatively and methadone postoperatively if the animals experienced pain, thereby ensuring excellent analgesia. In fact, one of the objectives of veterinary medicine is to provide adequate analgesia to help the patient not feel pain, and to move, eat, and sleep without discomfort, particularly in the first hours after the operation [17]. The use of local anaesthetics, despite its benefits, can cause side effects, including allergic reactions, sedation, respiratory depression, convulsions, hyperexcitability, tissue irrigation, and coma [18,19,20,21]. In this study, which included 20 cats, no side effects, complications, or mortality occurred in any of the patients.

The main result of this study was that the analgesia achieved in the LR group was more satisfactory than that in the SUF group. Multimodal anaesthesia, which was performed in the LR group, included drugs administered both systemically and regionally and is considered the most effective approach to providing pain relief [22]. This type of anaesthesia includes single drugs or drug combinations administered in varying dosages and routes and for various times; it is a less standardised approach but better suited to individual patients’ needs [23]. In particular, the LR group benefited from neuraxial anaesthesia, which included the administration of regional anaesthetic drugs in the epidural or subarachnoid space. This type of epidural anaesthesia blocks the motor, sensory, and autonomic systems; decreases the need for inhalants; and significantly reduces postoperative analgesic needs [24,25]. Most patients undergoing posterior quarter surgery involving the pelvic limbs, pelvis, or tail can benefit from the analgesia provided by neuraxial anaesthesia [26]. In addition to spinal administration, cats in the LR group benefited from the intercostal blockade. In particular, an intercostal block from T12 to T4 was performed because, due to the overlap of the innervation, it is necessary to block a minimum of three consecutive ribs to provide greater efficacy with this analgesia [27]. The advantage is that this blockage desensitises the tissues innervated between each rib, providing excellent analgesia without side effects or respiratory depression [9]. This study demonstrated that the use of loco-regional anaesthesia techniques was effective when using long-acting anaesthetics. In fact, compared to lidocaine and rupivacaine, the effect of levobupivacaine lasts approximately twice as long, guaranteeing effective analgesic coverage for at least 6 h after surgery [15,18,27]. Hence, levobupivacaine can minimise the surgical stress that inevitably emerges during and after invasive surgical procedures, such as radical mastectomy. This work demonstrates that spinal analgesia combined with intercostal analgesia provides excellent analgesia in patients undergoing unilateral mastectomy. In addition, in our study, all surgeries lasted approximately 50 min, and the analgesic block did not influence the duration of the surgery. In this study, it was difficult to choose the pain assessment system for cats. In fact, cats tend to hide pain as a protective mechanism [28]. Therefore, pain assessment in cats is complex [29,30]. Neuroendocrine tests can be performed to recognise pain, including measurements of norepinephrine, epinephrine, cortisol, and blood sugar levels. However, these neurochemicals are not specific to pain in cats and are affected by many factors. Behavioural changes are considered excellent indicators of pain in cats [31,32]. We used a behaviour rating system, rather than neuroendocrine tests, to assess the pain in cats. The UNESP-Botucatu pain scale [13,14] is widely used in cats and was chosen as the pain assessment scale in this study. The analgesic technique used in the LR group resulted in extremely low UNESP-Botucatu scores in the 8 h post-surgical assessment. Consequently, there was no routine administration of analgesics during that period. Similarly, the use of the pain scale made it possible to identify the precise time during which patients in the control group could have benefited from analgesic coverage, while avoiding unnecessary and counterproductive blind administrations. The UNESP-Botucatu scale, given the physical-behavioural aspects it takes into consideration, is useful and relatively easy to apply for the recognition and management of acute pain, thereby contributing to the improvement of overall wellbeing for veterinary patients.

## 5. Conclusions

This study demonstrated that the use of spinal anaesthesia with anaesthetic block of intercostal nerves guarantees long-lasting and better analgesic postoperative coverage compared to a CRI of sufentanyl, during unilateral mastectomy in cats.

## Figures and Tables

**Figure 1 animals-11-00887-f001:**
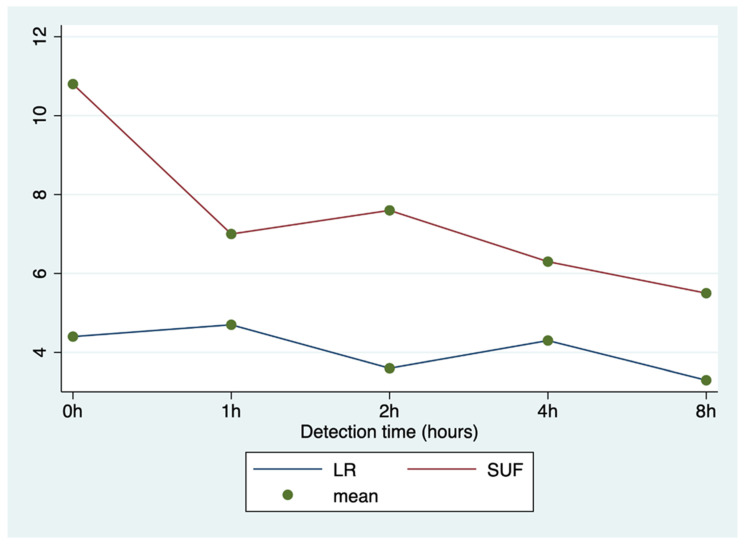
Average UNESP-Botucatu scores by group (LR vs. SUF) at different detection times.

**Table 1 animals-11-00887-t001:** Sample characteristics by group (LR vs. SUF).

Variable	Group		Total (*n* = 20)	*p*-Value
	LR (*n* = 10)	SUF (*n* = 10)		
Age (years)	13.0 ± 1.2 (11–15)	14.2 ± 2.0 (11–16)	13.6 ± 1.7 (11–16)	0.118
Weight (kg)	3.3 ± 0.5 (2–4)	3.1 ± 0.6 (2–4)	3.2 ± 0.5 (2–4)	0.341

Results are shown as mean ± standard deviation (range).

**Table 2 animals-11-00887-t002:** Methadone administration by group (LR vs. SUF).

Variable	LR (*n* = 10)	SUF (*n* = 10)	Total (*n* = 20)	*p*-Value
Drug administration, *n* (%)	4 (40.0)	10 (100.0)	14 (70.0)	0.011
Number of doses, *n* (%)				0.505
1	4/4 (100.0)	7/10 (70.0)	11/14 (78.6)	
2	0/4 (0.0)	3/10 (30.0)	3/14 (21.4)	
Administration dose 1, *n* (%)				0.203
0	1/4 (25.0)	7/10 (70.0)	8/14 (57.1)	
1	2/4 (50.0)	2/10 (20.0)	4/14 (28.6)	
2	0/4 (0.0)	1/10 (10.0)	1/14 (7.1)	
4	1/4 (25.0)	0/10 (0.0)	1/14 (7.1)	
Administration dose 2, *n* (%)				-
2	-	2/3 (66.7)	2/3 (66.7)	
4	-	1/3 (33.3)	1/3 (33.3)	

**Table 3 animals-11-00887-t003:** Analysis of the determinants of fentanyl administration in a multivariate logistic regression model.

Determinant	aOR	95% CI	*p*-Value
Weight	3.3	0.2–54.0	0.400
Age	0.2	0.1–0.9	0.043
Group (SUF vs. LR)	34.3	0.5–2340.9	0.100

Abbreviations: aOR, adjusted odds ratio; CI, confidence interval.

**Table 4 animals-11-00887-t004:** Analysis of the determinants of methadone administration in a multivariate logistic regression model.

Determinant	aOR	95% CI	*p*-Value
Weight	6.2	0.4–106.5	0.212
Age	1.1	0.3–3.8	0.830
Group (SUF vs. LR) *	1	-	0.100

Abbreviations: aOR, adjusted odds ratio; CI, confidence interval. * Predicted success perfectly.

**Table 5 animals-11-00887-t005:** Analysis of the determinants of the difference in UNESP-Botucatu scores between hours 8 and 0 in a multivariate linear regression model.

Determinant	Correlation Coefficient	95% CI	*p*-Value
Weight	1.6	−1.5 to 4.6	0.293
Age	−0.4	−1.4 to 0.6	0.435
Group (SUF vs. LR)	−3.4	−6.9 to 0.2	0.061

Abbreviations: CI, confidence interval.

## Data Availability

The data presented in this study are available on request from the corresponding author.

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
