# Peer review of "Use of Spinal Anaesthesia with Anaesthetic Block of Intercostal Nerves Compared to a Continuous Infusion of Sufentanyl to Improve Analgesia in Cats Undergoing Unilateral Mastectomy"

_animals, 2021, doi:10.3390/ani11030887_

Round 1

Reviewer 1 Report

The aim of the article titled “Effects of spinal anaesthesia with anaesthetic block of intercostal nerves to improve analgesia in cats undergoing unilateral mastectomy” by  Vincenzo Cicirelli, Pasquale Debidda, Nicola Maggio, Michele Caira, Daniela Mrenoshki, Giovani M Lacalandra, is to highlight the need to find new strategies to manage the pain in cats undergoing unilateral mastectomy. Despite the contents are interesting, in some places they are not well defined, therefore they needs more detailed clarification. Moreover, the paper needs an English language revision and to uniform the word style.

It is reconsidered after major revisions.

I would suggest to do the following changes:

Abstract

Line 13:  add the acronym CRI and in cats undergoing unilateral mastectomy

Line 17: add only the acronym CRI

Introduction

Line 38: explain because the postoperative pain is severe

Line 42-43: it needs to mention at least these two papers: Evaluation of tumescent local anesthesia in cats undergoing unilateral mastectomy. Clarissa MR Moreira et al., Vet Anaest Analg 2021; Effects of Surgical Wound Infiltration with Bupivacaine on Postoperative Analgesia in Cats Undergoing Bilateral Mastectomy. Özge Turna Yilmaz  et al., J Vet Med Sci 2014.

Line 46  add in cats undergoing unilateral mastectomy

Materials and Methods

  • there are no animal inclusion-exclusion criteria
  • there is no physical examination
  • there is no stadiation
  • there is no baseline
  • explain why you wanted to associate the spinal anaesthesia
  • explain better the effectorial sites of the spinal anaesthesia and the dilution of the drug administered
  • LINE 114: “lasted approximately 50 minutes”. It is appropriate specify a mean and SD
  • THERE IS NO NUM. OF AUTHORIZATION of Ethical Committee.
  • Line 130: D. Lgs 116/92: this law has been replaced from DIRECTIVE 2010/63/EUOF THE EUROPEAN PARLIAMENT (Italian DL 26/2014). It is considered a severe omission. Contemplate updating reference.

  • Data analysis: I DO NOT HAVE SUFFICIENT STATISTICAL SKILLS TO ASSESS THE STATISTICAL METHOD OF EVALUATED DATA

Discussion

Line 194-195: Explain this assertion adding more references: 1. The use of local anaesthetics, despite its benefits, can cause side effects including allergic reactions, sedation, respiratory depression, convulsions, hyperexcitability, tissue irrigation, and coma. I suggest citing or considering the following papers: 2. Local and regional anaesthesia in dogs and cats: Overview of concepts and drugs (Part 1), Grubb and Lobprise, 2020, Vet Med Sci; 3. Local and regional anaesthesia in dogs and cats: Descriptions of specific local and regional techniques (Part 2), Grubb and Lobprise, 2020, Vet Med Sci; 4.

Line 204: Add and comment the following reference: Combined spinal and general anaesthesia in 58 cats undergoing various surgical procedures: description of technique and retrospective peri-operative evaluation (Sarotti et al, 2018, J Feline Med and Surg); Small animal regional anesthesia and analgesia, Campoy and Read, 2013.

References

Add and comment the following most recent references than those mentioned:

  1. Cassali et al: Consensus for the diagnosis, prognosis and treatment of feline mammary tumors. 2018, Brazilian Journal of Veterinary Research and Animal Science 55(2):1-17.9
  2. Wypij et al: Malignant mammary tumors: Biologic behavior, prognostic factors, and therapeutic approach in cats. 2006, Veterinary Medicine
  3. De Campos et al: Use of Surgery and Carboplatin in Feline Malignant Mammary Gland Neoplasms with Advanced Clinical Staging. 2014, In Vivo.
  4. Steagall et al: Feline anesthesia and pain management. 2017

Reference 27:  from 2016 there is the new edition….

Author Response

Reviewer #1

The aim of the article titled “Effects of spinal anaesthesia with anaesthetic block of intercostal nerves to improve analgesia in cats undergoing unilateral mastectomy” by  Vincenzo Cicirelli, Pasquale Debidda, Nicola Maggio, Michele Caira, Daniela Mrenoshki, Giovani M Lacalandra, is to highlight the need to find new strategies to manage the pain in cats undergoing unilateral mastectomy. Despite the contents are interesting, in some places they are not well defined, therefore they needs more detailed clarification. Moreover, the paper needs an English language revision and to uniform the word style. It is reconsidered after major revisions.

Thank you for sharing your valuable feedback. In light of the journal reviewers' comments, I have assessed the edited file and found that there were indeed some language-related errors that were not resolved during the edit. For this reasons, I improved the English language of the manuscript with the help of an English native speaker with a long history of correcting scientific manuscripts for submission. Also, my manuscript was edit by Elsevier Language Editing Services.

I would suggest to do the following changes:

 Abstract

Line 13:  add the acronym CRI and in cats undergoing unilateral mastectomy

Line 17: add only the acronym CRI

Ok, thanks.

Introduction

Line 38: explain because the postoperative pain is severe

Ok, done.

Line 42-43: it needs to mention at least these two papers: Evaluation of tumescent local anesthesia in cats undergoing unilateral mastectomy. Clarissa MR Moreira et al., Vet Anaest Analg 2021; Effects of Surgical Wound Infiltration with Bupivacaine on Postoperative Analgesia in Cats Undergoing Bilateral Mastectomy. Özge Turna Yilmaz  et al., J Vet Med Sci 2014.

Ok, done.

Line 46  add in cats undergoing unilateral mastectomy

Ok, thanks.

  • Materials and Methods

there are no animal inclusion-exclusion criteria

there is no physical examination

there is no stadiation

there is no baseline

explain why you wanted to associate the spinal anaesthesia

explain better the effectorial sites of the spinal anaesthesia and the dilution of the drug administered

.LINE 114: “lasted approximately 50 minutes”. It is appropriate specify a mean and SD

THERE IS NO NUM. OF AUTHORIZATION of Ethical Committee.

Line 130: D. Lgs 116/92: this law has been replaced from DIRECTIVE 2010/63/EUOF THE EUROPEAN PARLIAMENT (Italian DL 26/2014). It is considered a severe omission. Contemplate updating reference.

I add this informations in the manuscript

Data analysis: I DO NOT HAVE SUFFICIENT STATISTICAL SKILLS TO ASSESS THE STATISTICAL METHOD OF EVALUATED DATA

Discussion

Line 194-195: Explain this assertion adding more references: 1. The use of local anaesthetics, despite its benefits, can cause side effects including allergic reactions, sedation, respiratory depression, convulsions, hyperexcitability, tissue irrigation, and coma. I suggest citing or considering the following papers: 2. Local and regional anaesthesia in dogs and cats: Overview of concepts and drugs (Part 1), Grubb and Lobprise, 2020, Vet Med Sci; 3. Local and regional anaesthesia in dogs and cats: Descriptions of specific local and regional techniques (Part 2), Grubb and Lobprise, 2020, Vet Med Sci; 4.

Line 204: Add and comment the following reference: Combined spinal and general anaesthesia in 58 cats undergoing various surgical procedures: description of technique and retrospective peri-operative evaluation (Sarotti et al, 2018, J Feline Med and Surg); Small animal regional anesthesia and analgesia, Campoy and Read, 2013.

Ok.

References

Add and comment the following most recent references than those mentioned:

  1. Cassali et al: Consensus for the diagnosis, prognosis and treatment of feline mammary tumors. 2018, Brazilian Journal of Veterinary Research and Animal Science 55(2):1-17.9
  2. Wypij et al: Malignant mammary tumors: Biologic behavior, prognostic factors, and therapeutic approach in cats. 2006, Veterinary Medicine
  3. De Campos et al: Use of Surgery and Carboplatin in Feline Malignant Mammary Gland Neoplasms with Advanced Clinical Staging. 2014, In Vivo.
  4. Steagall et al: Feline anesthesia and pain management. 2017

Reference 27:  from 2016 there is the new edition….

Ok, done. Thanks a lot.

Reviewer 2 Report

Dear authors,

Thank you for the opportunity to review your manuscript. I think the manuscript needs to be greatly improved before it could be considered for publication. The use of English should be completely revised and the style improved. Likewise, an effort should be made to improve the writing and structure of the entire text.

I believe that an extensive and up-to-date literature review is necessary to rewrite the introduction and discussion entirely.

Below I share some of my suggestions and comments.

Kind regards,

***********

Title

Authors should reformulate the title of the article.

In this manuscript, the analgesic efficacy of spinal anesthesia and intercostal block have been evaluated compared to a continuous infusion of sufentanyl. No other effects of the locoregional techniques used are mentioned.

Keywords

Keywords that are not included in the title increase the reader's ability to search for the article. I would suggest to modify them.

Simple summary:

It is necessary to follow the rules set out in ‘Instructions for Authors’: ‘The simple summary consists of no more than 200 words in one paragraph and contains a clear statement of the problem addressed, the aims and objectives, pertinent results, conclusions from the study and how they will be valuable to society.’

This part of the paper is totally disorganized. It needs to be entirely rewritten and summarized.

Line 9: Unilateral instead unilaterar

Line 14: It is made a reference to the SUF group having received fentanyl instead of sufentanyl.

Lines 16-18: This sentence seems to be the conclusion of the study and the authors put it in the middle of the simple summary.

Line 19: This sentence should be at the beginning of the simple summary, it is part of the introduction

Lines 20-22: Hypotheses are mixed with conclusions from the study.

Abstract:

The abstract should be a total of about 200 words maximum.

Line 24: Unilateral instead unilaterar

Line 31: It is made a reference to the SUF group having received fentanyl instead of sufentanyl.

Introduction:

In my opinion, the introduction is very brief, even shorter than the simple summary or the abstract.

Line 56. The authors mention that there are few studies on spinal anesthesia in feline mastectomies but they do not even cite any of them.

Other locoregional techniques have been used in these interventions in cats that are not mentioned as tumescent anesthesia or infiltration of wounds ...

The authors should carry out an exhaustive review of the bibliography and completely rewrite the introduction, adequately establishing the state of the art of the subject.

Material & Methods:

Line 63: Why were 20 cats selected? Has an evaluation of the sample size been done?

Line 68: LR has not been previously defined

Line 71: SUF has not been previously defined

Line 81-82: This sentence sounds quite weird to me.

Line 84: It is not clear what percentage of Et Sevo the cats were kept. (Et Sevo 2, 3/2.7%) The commercial name of sevoflurane is not well referenced

Line 84-89: Only the baseline data record is discussed before the incision. How was it done later? How often?

Lines 89-92: I consider that this sentence must be included in 2.3 Surgery procedure, not in 2.1 Pre-surgery procedure.

Line 93: I miss some bibliographic references when describing the technique of spinal anesthesia and intercostal block.

Line 130: What do the authors mean by the evaluation criteria

Line 138: Rescue analgesia should be established. Which one was used?

Results

Line 172: Maybe, intraoperative sounds better than intra-surgery.

Line 180: I understand that methadone is postoperative rescue analgesia? It was not explained in M&M (line 138)

Discussion

Lines 201-203: The sentence must be rephrased, it sounds strange. The verb is missing.

Line 227: paralyses or desensitized?

Line 229: This information must be included in M&M when the intercostal block is explained. It is the first time for the lector to know in which intercostal spaces the block has been performed.

Line 235: Levobupivacaine instead of bupivacaine?

Conclusion

Lines 256-260: The conclusion seems too ambitious to me and should be reworded. It is not possible to speak of locoregional techniques in general, only to refer to the one used in the study. Furthermore, analgesic efficacy has only been evaluated based on intraoperative and postoperative opioid needs, nothing else. Surgical stress has not been evaluated.

Lines 261-264: In my opinion, this is not a conclusion of this study; I would eliminate it but if it is put it should be in the discussion.

Author Response

Reviewer #2

Dear authors,

Thank you for the opportunity to review your manuscript. I think the manuscript needs to be greatly improved before it could be considered for publication. The use of English should be completely revised and the style improved. Likewise, an effort should be made to improve the writing and structure of the entire text.

I believe that an extensive and up-to-date literature review is necessary to rewrite the introduction and discussion entirely.

Below I share some of my suggestions and comments.

Kind regards,

Thank you for sharing your valuable feedback. In light of the journal reviewers' comments, I have assessed the edited file and found that there were indeed some language-related errors that were not resolved during the edit. For this reasons, I improved the English language of the manuscript with the help of an English native speaker with a long history of correcting scientific manuscripts for submission. Also, my manuscript was edit by Elsevier Language Editing Services.

***********

Title

Authors should reformulate the title of the article.

In this manuscript, the analgesic efficacy of spinal anesthesia and intercostal block have been evaluated compared to a continuous infusion of sufentanyl. No other effects of the locoregional techniques used are mentioned.

Thanks for the advice. I changed the title considering your suggestions.

Keywords

Keywords that are not included in the title increase the reader's ability to search for the article. I would suggest to modify them.

Dear reviewer, thank you for your comments which are useful in oreder to improve the manuscript.

I changed some keywords considering your suggestions.

Simple summary:

It is necessary to follow the rules set out in ‘Instructions for Authors’: ‘The simple summary consists of no more than 200 words in one paragraph and contains a clear statement of the problem addressed, the aims and objectives, pertinent results, conclusions from the study and how they will be valuable to society.’

This part of the paper is totally disorganized. It needs to be entirely rewritten and summarized.

Line 9: Unilateral instead unilaterar

Line 14: It is made a reference to the SUF group having received fentanyl instead of sufentanyl.

Lines 16-18: This sentence seems to be the conclusion of the study and the authors put it in the middle of the simple summary.

Line 19: This sentence should be at the beginning of the simple summary, it is part of the introduction

Lines 20-22: Hypotheses are mixed with conclusions from the study.

I rewrote the simple summary considering “Instructions for Authors” and your advice.

Abstract:

The abstract should be a total of about 200 words maximum.

Line 24: Unilateral instead unilaterar

Line 31: It is made a reference to the SUF group having received fentanyl instead of sufentanyl.

I reduced the abstract (200 words), focusing on the project presented.

Introduction:

In my opinion, the introduction is very brief, even shorter than the simple summary or the abstract.

Line 56. The authors mention that there are few studies on spinal anesthesia in feline mastectomies but they do not even cite any of them.

Other locoregional techniques have been used in these interventions in cats that are not mentioned as tumescent anesthesia or infiltration of wounds ...

The authors should carry out an exhaustive review of the bibliography and completely rewrite the introduction, adequately establishing the state of the art of the subject.

I rewrote the introduction focusing on the project presented and adding recent bibliographic references, establishing the state of the art of the subject.

Material & Methods:

Line 63: Why were 20 cats selected? Has an evaluation of the sample size been done?

Line 68: LR has not been previously defined

Line 71: SUF has not been previously defined

Line 81-82: This sentence sounds quite weird to me.

Line 84: It is not clear what percentage of Et Sevo the cats were kept. (Et Sevo 2, 3/2.7%) The commercial name of sevoflurane is not well referenced

Line 84-89: Only the baseline data record is discussed before the incision. How was it done later? How often?

Lines 89-92: I consider that this sentence must be included in 2.3 Surgery procedure, not in 2.1 Pre-surgery procedure.

Line 93: I miss some bibliographic references when describing the technique of spinal anesthesia and intercostal block.

Line 130: What do the authors mean by the evaluation criteria

In the event of an increased heart rate or blood pressure (>30% compared to pre-incision values) during the procedure in response to surgical pain in both groups, a bolus of fentanyl was administered [13] intravenously at 2 µg/kg (Fentadon®; Eurovet Animal Health BV).

Line 138: Rescue analgesia should be established. Which one was used?

Ok, thanks. The section of Material and methods was revised, considering your suggest.

Results

Line 172: Maybe, intraoperative sounds better than intra-surgery.

Line 180: I understand that methadone is postoperative rescue analgesia? It was not explained in M&M (line 138)

Ok, thanks. The section of Results was revised, considering your suggest.

Discussion

Lines 201-203: The sentence must be rephrased, it sounds strange. The verb is missing.

Line 227: paralyses or desensitized?

Of course “Desensitized” is better. Thanks.

Line 229: This information must be included in M&M when the intercostal block is explained. It is the first time for the lector to know in which intercostal spaces the block has been performed.

Ok, done.

Line 235: Levobupivacaine instead of bupivacaine?

Levobupivacaine.

Conclusion

Lines 256-260: The conclusion seems too ambitious to me and should be reworded. It is not possible to speak of locoregional techniques in general, only to refer to the one used in the study. Furthermore, analgesic efficacy has only been evaluated based on intraoperative and postoperative opioid needs, nothing else. Surgical stress has not been evaluated.

Lines 261-264: In my opinion, this is not a conclusion of this study; I would eliminate it but if it is put it should be in the discussion.

I deleted that sentence following your advice.

Round 2

Reviewer 1 Report

Simple summary:

line 9: remove the first paragraph because it is repeated in the abstract.

line 14: add in cat undergoing unilateral mastectomy

Materials and methods

line 143: the reference n. 11 is not correct, because two references have been put together (one is a publication and one is a book, see the paragraph of references). I suggest you to remove the number of reference at this point (line 143) because not relevant, and to mention it in the Discussion (Diego Sarotti, Andrea Cattai, Paolo Franci: Combined spinal and general anaesthesia in 58 cats undergoing various surgical procedures: description of technique and retrospective perioperative evaluation. Journal of Feline Medicine and Surgery 1–7, 2018), because it represents an important paper that explain the advantages and disadvantages of a spinal anaesthesia in cats

References

Add this reference: Otero PE and Campoy L. Epidural and spinal anesthesia. In Campoy L and Read M. Small animals and regional anesthesia and analgesia. New York: Wiley-Blackwell, 2013, pp 227–231.

Author Response

Simple summary:

line 9: remove the first paragraph because it is repeated in the abstract.

line 14: add in cat undergoing unilateral mastectomy

Materials and methods

line 143: the reference n. 11 is not correct, because two references have been put together (one is a publication and one is a book, see the paragraph of references). I suggest you to remove the number of reference at this point (line 143) because not relevant, and to mention it in the Discussion (Diego Sarotti, Andrea Cattai, Paolo Franci: Combined spinal and general anaesthesia in 58 cats undergoing various surgical procedures: description of technique and retrospective perioperative evaluation. Journal of Feline Medicine and Surgery 1–7, 2018), because it represents an important paper that explain the advantages and disadvantages of a spinal anaesthesia in cats

References

Add this reference: Otero PE and Campoy L. Epidural and spinal anesthesia. In Campoy L and Read M. Small animals and regional anesthesia and analgesia. New York: Wiley-Blackwell, 2013, pp 227–231.

Dear reviewer, thank you for your comments which are useful in oreder to improve the manuscript. I made the suggested corrections and added the bibliographical references you indicated.

Reviewer 2 Report

Dear authors,

Thank you for the work done to edit the manuscript. I believe that there are still points that are not clarified and that need answer or correction.

Best regards,

***************

Title

In this manuscript, the analgesic efficacy of spinal anaesthesia and intercostal block have been evaluated compared to a continuous infusion of sufentanil. I think it would be more appropriate:

Use of spinal anaesthesia with anaesthetic block of intercostal nerves compared to a continuous infusion of sufentanil to improve analgesia in cats undergoing unilateral mastectomy

Simple summary:

Line 13: I would substitute "… with the constant-rate infusion…" for "… with a constant-rate infusion…"

Lines 14-17: In my opinion, the authors should not state this based on the study carried out. They should only refer to the techniques used in the case of feline mastectomies. Also, as I said in the previous review, in my opinion, surgical stress has not been evaluated, has it? If it has been, the authors should explain it in M & M and Results.

Abstract

Lines 29 and 30: LR and SUF have not been previously defined

Line 31:

“…which received a CRI of fentanyl”. I think it should be sufentanil

Line 50: I think you should be consistent with the terms, post-operative or postoperative, but the same throughout the entire manuscript.

Introduction

I'm sorry to be reiterative but in my opinion, the introduction still does not provide a good review of the subject to the readers. Other locoregional techniques have been used in these interventions in cats that are not mentioned as tumescent anaesthesia or infiltration of wounds ... Only a reference (#7) has been included without making any mention. Nothing is explained about the use of intercostal nerve blocks neither in mastectomies nor on the use of sufentanil infusions in cats. On the contrary, a quarter of the introduction is the assessment of pain using the UNESP-Botucatu scale.

I sincerely believe that it should be restructured to provide the reader with all the relevant information on the subject before raising the study.

Line 56. The authors mention that there are few studies on spinal anaesthesia in feline mastectomies but they do not even cite any of them.

Other locoregional techniques have been used in these interventions in cats that are not mentioned as tumescent anaesthesia or infiltration of wounds...

The authors should carry out an exhaustive review of the bibliography and completely rewrite the introduction, adequately establishing the state of the art of the subject.

Material & Methods:

Line 63: Why were 20 cats selected? Has an evaluation of the sample size been done?

Line 91: I think it should be sufentanil instead of fentanyl

Line 105-111: Only the baseline data record is discussed before the incision. How was it done later? How often was the data of monitoring registered?

Line 143: Until when was the CRI of sufentanil maintained? Was it stopped at the end of the intervention or was it maintained while postoperative pain was assessed until 8 hours? Please state that.

Results

Perhaps in table 1, the abbreviations LR and SUF should be defined in the table legend.

Discussion

Lines 253-254: Attending to your results, at the intraoperative level, no significant differences were found between the two groups with respect to the administration of fentanyl after the detection of a nociceptive response.

Obviously, there are differences between both groups in terms of rescue analgesia with methadone, but it does not seem to me that having to rescue 4 out of 10 cats after ALI know how to say that the anaesthesia provided in the LR group is adequate in this type of intervention.

Lines 261-266: This paragraph should be deleted as it has nothing to do with what is being discussed. It would be better to discuss the results of the study with those of Moreira et al. 2021 and with Yilmaz et al 2014 (doi: 10.1292 / jvms.14-0112).

Line 277: bupivacaine?

Lines 281-283: I recommend deleting this sentence or rephrasing it because this is not supported by the results of the study.

Conclusion

In my opinion, this conclusion cannot be drawn from the work. I believe that what is demonstrated is that the animals in the LR group required fewer analgesic interventions at the postoperative level than those in the SUF group, but it must be remembered that almost 1 out of every 2 cats required analgesic rescue. I don’t think it could be considered that in this study spinal anaesthesia with intercostal nerve block provides excellent analgesia.

References

Please check the bibliographic references because some of them need to be edited (7, 11,..)

Author Response

Comments and Suggestions for Authors

Dear authors,

Thank you for the work done to edit the manuscript. I believe that there are still points that are not clarified and that need answer or correction.

Best regards,

 Dear reviewer, thank you for your comments which are useful in oreder to improve the manuscript. As suggested by you, I expanded the introduction, focusing on the spinal anaesthesia with anaesthetic block of intercostal nerves compared to a continuous infusion of sufentanyl to improve analgesia in cats undergoing unilateral mastectomy

Title

In this manuscript, the analgesic efficacy of spinal anaesthesia and intercostal block have been evaluated compared to a continuous infusion of sufentanil. I think it would be more appropriate:

Use of spinal anaesthesia with anaesthetic block of intercostal nerves compared to a continuous infusion of sufentanil to improve analgesia in cats undergoing unilateral mastectomy

Ok, thanks. The new title of the paper is: “Use of spinal anaesthesia with anaesthetic block of intercostal nerves compared to a continuous infusion of sufentanil to improve analgesia in cats undergoing unilateral mastectomy”

Simple summary:

Line 13: I would substitute "… with the constant-rate infusion…" for "… with a constant-rate infusion…"

Lines 14-17: In my opinion, the authors should not state this based on the study carried out. They should only refer to the techniques used in the case of feline mastectomies. Also, as I said in the previous review, in my opinion, surgical stress has not been evaluated, has it? If it has been, the authors should explain it in M & M and Results.

I corrected in according to your suggestions. In additio, in this paper surgical stress has not been evaluated; however in surgery, in the event of an increased heart rate or blood pressure (>30% compared to pre-incision values), a bolus of fentanyl was administered intravenously at 2 µg/kg.

Abstract

Lines 29 and 30: LR and SUF have not been previously defined

Ok, I have better defined the 2 groups.

Line 31:

“…which received a CRI of fentanyl”. I think it should be sufentanil

Ok.

Line 50: I think you should be consistent with the terms, post-operative or postoperative, but the same throughout the entire manuscript.

Ok, done

Introduction

I'm sorry to be reiterative but in my opinion, the introduction still does not provide a good review of the subject to the readers. Other locoregional techniques have been used in these interventions in cats that are not mentioned as tumescent anaesthesia or infiltration of wounds ... Only a reference (#7) has been included without making any mention. Nothing is explained about the use of intercostal nerve blocks neither in mastectomies nor on the use of sufentanil infusions in cats. On the contrary, a quarter of the introduction is the assessment of pain using the UNESP-Botucatu scale.

I sincerely believe that it should be restructured to provide the reader with all the relevant information on the subject before raising the study.

Line 56. The authors mention that there are few studies on spinal anaesthesia in feline mastectomies but they do not even cite any of them.

Other locoregional techniques have been used in these interventions in cats that are not mentioned as tumescent anaesthesia or infiltration of wounds...

The authors should carry out an exhaustive review of the bibliography and completely rewrite the introduction, adequately establishing the state of the art of the subject.

 I restructured the introduction focusing on the project presented and adding recent bibliographic references about “tumescent anaesthesia” and “infiltration of wounds” local anaesthetics, establishing the state of the art of the subject.

Material & Methods:

Line 63: Why were 20 cats selected? Has an evaluation of the sample size been done?

20 cats were selected because it is the minimum number to carry out a good statistical study.

Line 91: I think it should be sufentanil instead of fentanyl

ok

Line 105-111: Only the baseline data record is discussed before the incision. How was it done later? How often was the data of monitoring registered?

I explained it better with this sentence: From this point, continuously throughout the surgery, instrumental monitoring of the following vital parameters was started (pre-incision values): heart rate, electrocardiographic trace, pulse oximetry, carbon dioxide concentration at the end of expiration, percentage of sevoflurane at the end of expiration, and non-invasive blood pressure and body temperature (GE-Datex Ohmeda Carestation 620 Anaesthesia Cart, GE-Datex Ohmeda B 450 Monitor).

Line 143: Until when was the CRI of sufentanil maintained? Was it stopped at the end of the intervention or was it maintained while postoperative pain was assessed until 8 hours? Please state that.

I added this sentence: For the cats in the SUF group, systemic analgesia was performed with administration of sufentanyl citrate (50 µg/mL) with a bolus loading of 1 µg/kg and maintenance of 1.5 µg/kg/h at a constant-rate intravenous infusion until the end of the surgery.

Results

Perhaps in table 1, the abbreviations LR and SUF should be defined in the table legend.

ok

Discussion

Lines 253-254: Attending to your results, at the intraoperative level, no significant differences were found between the two groups with respect to the administration of fentanyl after the detection of a nociceptive response.

Obviously, there are differences between both groups in terms of rescue analgesia with methadone, but it does not seem to me that having to rescue 4 out of 10 cats after ALI know how to say that the anaesthesia provided in the LR group is adequate in this type of intervention.

Normally, in daily practice, in many mastectomy in the cat, rescue analgesia is used. This also happens using excellent protocols such as sufentanyl CRI, because it is a very painful surgery.

In our protocol we had a good result, having only used rescue analgesia in 4 out of 10 patients, compared with 10 patients of the sufentanyl CRI group.

Lines 261-266: This paragraph should be deleted as it has nothing to do with what is being discussed. It would be better to discuss the results of the study with those of Moreira et al. 2021 and with Yilmaz et al 2014 (doi: 10.1292 / jvms.14-0112).

Ok, thanks.

Line 277: bupivacaine?

Changed.

Lines 281-283: I recommend deleting this sentence or rephrasing it because this is not supported by the results of the study.

Ok.

Conclusion

In my opinion, this conclusion cannot be drawn from the work. I believe that what is demonstrated is that the animals in the LR group required fewer analgesic interventions at the postoperative level than those in the SUF group, but it must be remembered that almost 1 out of every 2 cats required analgesic rescue. I don’t think it could be considered that in this study spinal anaesthesia with intercostal nerve block provides excellent analgesia.

I rephrased the conclusion as follows:

This study demonstrated that the use of spinal anaesthesia with an anaesthetic block of intercostal nerves guarantees long-lasting and better analgesic postoperative coverage compared to a CRI of sufentanyl, during unilateral mastectomy in cats.

References

Please check the bibliographic references because some of them need to be edited (7, 11,..)

Ok, thanks
